# Emotional Dysregulation, Temperament and Lifetime Suicidal Ideation among Youths with Mood Disorders

**DOI:** 10.3390/jpm11090865

**Published:** 2021-08-30

**Authors:** Delfina Janiri, Lorenzo Moccia, Eliana Conte, Laura Palumbo, Daniela Pia Rosaria Chieffo, Giulia Fredda, Renato Maria Menichincheri, Andrea Balbi, Georgios D. Kotzalidis, Gabriele Sani, Luigi Janiri

**Affiliations:** 1Department of Neuroscience, Section of Psychiatry, Università Cattolica del Sacro Cuore, 00168 Rome, Italy; delfina.janiri@uniroma1.it (D.J.); lorenzo.moccia@unicatt.it (L.M.); eliana.conte@aslroma3.it (E.C.); laura.palumbo01@icatt.it (L.P.); luigi.janiri@unicatt.it (L.J.); 2Department of Human Neuroscience, Sapienza University of Rome, 00185 Rome, Italy; 3Department of Psychiatry, Fondazione Policlinico Universitario Agostino Gemelli IRCCS, 00168 Rome, Italy; 4Early Intervention Unit, ASL Roma 3, 00152 Rome, Italy; renatomaria.menichincheri@aslroma3.it (R.M.M.); andrea.balbi@aslroma3.it (A.B.); 5Clinical Psychology Unit, Fondazione Policlinico Gemelli IRCCS, 00168 Rome, Italy; danielapiarosaria.chieffo@policlinicogemelli.it (D.P.R.C.); giulia.fredda@guest.policlinicogemelli.it (G.F.); 6Department of Neurosciences, Mental Health and Sensory Organs (NESMOS), Sapienza University of Rome, 00189 Rome, Italy; giorgio.kotzalidis@uniroma1.it

**Keywords:** emotional dysregulation, affective temperaments, depressive disorders, bipolar disorders, youth

## Abstract

*Background*: Psychopathological dimensions contributing to suicidal ideation in young age are poorly understood. We aimed to investigate the involvement of emotional dysregulation and temperament in suicide risk in a sample of accurately selected young patients with mood disorders and a matched sample of healthy controls (HC). *Methods*: We assessed 50 young patients (aged 14–25 years) with DSM-5 bipolar or depressive disorders for clinical and psychopathological characteristics and 82 age and sex, educational level, and smoking habits-matched HC. Emotional dysregulation and temperament were assessed using the Difficulties in Emotion Regulation Scale (DERS) and the Temperament Evaluation of Memphis, Pisa, Paris and San Diego-Autoquestionnaire (TEMPS-A-39), respectively. We tested their associations with suicidal ideation, using standard univariate/bivariate methods, preceded by overall multivariate analysis. *Results*: In the group of patients, 24 (48%) reported lifetime suicide ideation (LSI). Patients with LSI scored higher on emotional dysregulation (*p* < 0.001) and cyclothymic (*p* < 0.001), irritable (*p* = 0.01), and hyperthymic temperaments (*p* = 0.003) than HC. Patients with LSI specifically presented with more emotional dysregulation (*p* < 0.001) and cyclothymic temperament (*p* = 0.001), than patients without LSI (N = 26). *Conclusions*: Temperamental features, in particular cyclothymic temperament, and emotion dysregulation may represent independent factors for increased vulnerability to lifetime suicidal ideation in young adults with mood disorders.

## 1. Introduction

In adolescents and young adults, suicide rates have been steadily increasing over the last decade [1]. The number of hospitalizations and emergency visits for suicidal ideation and suicide attempts in this age group has doubled over time [2]. Suicidal ideas in young age predict adult psychiatric morbidity and may serve as a marker of vulnerability to psychopathology [3]. Prompt identification of suicidal ideation in youth allows intervention planning aiming at better functional outcomes in mental health [4].

Emotion dysregulation is conceptualized as difficulty in several areas, including the ability to monitor and evaluate emotional experiences, adapt to their intensity and duration, and modulate emotional reactions in order to meet situational demands [5]. It could also be defined as difficulty in regulating the rapid oscillations of intense affects [6]. Emotional dysregulation is associated with increased psychiatric morbidity, particularly mood disorders. The DSM-IV had grouped bipolar and depressive disorders under the heading of mood disorders, but the DSM-5 has split them in two different categories. Nevertheless, both major depressive and bipolar disorders present with emotional dysregulation, which may be considered a shared feature among mood disorders [7,8]. Emotional dysregulation may impact the clinical course of both major depressive and bipolar disorders. In particular, it is shown to constitute a biological determinant of suicide risk in both adults [9] and adolescents [10]. 

Temperament identifies stable, early-appearing characteristics in behavioral tendencies that have a constitutional and biological basis. Premorbid affective temperament types refer to individual activity levels, rhythms, mood and related cognitions, [11] and have an important role in the clinical evolution of mood disorders [12]. In particular, differences in temperament traits are differently associated with suicide risk in adult patients with mood disorders [13]. 

In light of the above observations, we decided to investigate whether there is a specific relationship between emotional dysregulation, temperament, and suicide risk in a sample of accurately selected young patients with a bipolar or a depressive disorder, and a matched sample of healthy individuals. 

## 2. Material and Methods 

### 2.1. Participants

We consecutively assessed 50 young outpatients who had been diagnosed with a DSM-5 [14] bipolar (N = 21) or depressive disorder (N = 29). Patients were enrolled at the Early Intervention for Mood Disorders Unit at Fondazione Policlinico Universitario Agostino Gemelli IRCCS in Rome, Italy. Patients were screened by trained staff for DSM-5 disorders, and clinical diagnoses were confirmed, using the Structured Clinical Interview for DSM-5–Research Version [15]. In addition to a diagnosis of mood disorder, inclusion criteria were as follows: (i) age between 14 and 25 years, (ii) stable phase of illness according to psychometric evaluation (Hamilton Depression Rating Scale, HAM-D  ≤  7; Young Mania Rating Scale  ≤  12), (iii) fluency in Italian, and (iv) at least five years of school education. Exclusion criteria wereas follows: (i) traumatic head injury with loss of consciousness; (ii) lifetime history of major medical or neurological disorders; (iii) suspected cognitive impairment based on a Mini-Mental State Examination (MMSE) [16] score lower than 24; (iv) recent (past six weeks) changes in any psychotropic medication; (v) current use of stimulant medications; and (vi) a history of psychosis unrelated to the primary mood disorder. We also recruited 82 healthy controls (HC), matched for age, sex, smoking status, and educational level, from the same geographical area. All HC were screened for current or lifetime history of DSM-5 disorders. For the aims of this study, they were also interviewed to determine their suicidal behavior potential; not one of them reported lifetime suicidal behavior. Participants were interviewed to assess whether any first-degree relative was affected by mood disorders or schizophrenia. If they had a positive family history, they were excluded. Other exclusion criteria were the same as those for the patient group. The study was approved and undertaken in accordance with the guidelines of the Fondazione Policlinico Universitario Agostino Gemelli Ethics Committee and in accordance with the Principles of Human Rights, as adopted by the World Medical Association at the 18th WMA General Assembly, Helsinki, Finland, June 1964 and subsequently amended at the 64th WMA General Assembly, Fortaleza, Brazil, October 2013. All participants gave their written informed consent to participate in the study after they had received a complete explanation of the study procedures.

### 2.2. Assessment

To assess deficits in emotion regulation, we used the Difficulties in Emotion Regulation Scale (DERS) [5], a 36-item self-report measure that assesses individuals’ typical levels of emotion dysregulation. Participants rate each item, using a 5-point Likert-type scale (ranging from 1  =  almost never, to 5  =  almost always). Higher scores indicate greater difficulties regulating emotions. In prior studies, the DERS demonstrated convergent validity with other established measures of emotion dysregulation, good test-retest reliability, excellent internal consistency and adequate predictive validity of several behavioral outcomes associated with emotion dysregulation [5].

Affective temperaments (cyclothymic, depressive, irritable, hyperthymic, and anxious) were assessed through the short, 39-item version of the validated Italian Temperament Evaluation of Memphis, Pisa, Paris and San Diego-Autoquestionnaire (TEMPS-A-39) [17]. This instrument is widely used in research and has demonstrated good psychometric properties and optimal factor structure [18].

Clinical characteristics were collected during a clinical interview. Lifetime suicidal ideation was assessed with a semi-structured questionnaire consisting of two parts, one related to the past 6 months, and the other, lifetime. Each part included two questions: (1) “Have you ever seriously thought about committing suicide?” (2) “Have you ever made a plan for committing suicide?” Respondents had to answer only “Yes” or “No”. The semi-structured questionnaire has not been yet validated, but it was already used previously by our group [11].

### 2.3. Statistical Analyses

We compared the three groups’ (i.e., patients with and without lifetime suicidal ideation, and HC) sociodemographic and clinical characteristics on the basis of the chi-squared (*χ*^2^) test for nominal variables and one-way analysis of variance (ANOVA1way) followed by post hoc Bonferroni tests for continuous variables and by pairwise post hoc analyses for nominal variables.

For the aims of this study, we focused on the distribution patterns of temperament and emotional dysregulation in the three groups. Accordingly, we conducted a series of one-way ANOVAs, followed by Bonferroni post hoc tests, to compare means among groups. The level of significance was set at *p* < 0.05 for the ANOVA comparative measurements. To minimize the likelihood of type I errors, the ANOVAs were preceded by overall multivariate analysis of variance (MANOVA) using all of the continuous variables considered in each of the analyses as dependent variables.

## 3. Results

The sociodemographic and clinical characteristics of the sample are shown in Table 1. In the total group of the 50 mood disorder patients, 24 (48%) reported lifetime suicide ideation (LSI). Regarding clinical characteristics, LSI patients reported more family history of psychiatric disorders (83.3%), more use of lithium (45.8%) and antipsychotic medications (50.0%) than patients without LSI (NoLSI) (Table 1). Furthermore, in the suicidal ideation group, most participants (70.8%) reported psychotherapy treatment (Table 1). There were no differences in belonging to the LSI or NoLSI groups among the diagnoses.

Regarding the distribution patterns of temperament and emotional dysregulation, a preliminary MANOVA revealed a significant global effect (Wilks’ Lambda = 0.56, F = 6.73, df = 12, *p* < 0.0001) of all the variables of interest on the three groups (i.e., patients with and without lifetime suicidal ideation, and HC). Factorial ANOVAs indicated a main effect of diagnosis on emotional dysregulation, cyclothymic, irritable, and hyperthymic temperaments (Table 2). In particular, a series of Bonferroni post hoc tests clarified that LSI patients scored higher on emotional dysregulation and cyclothymic, irritable, and hyperthymic temperaments than HC, whereas NoLSI patients scored higher than HC only on the irritable temperament. At the direct comparison, patients with and without LSI differed in emotional dysregulation and cyclothymic temperament. In particular, patients with LSI presented with higher emotional dysregulation and higher endorsement of the cyclothymic temperament. 

## 4. Discussion

Suicide is the second leading cause of death in young adults and adolescents with psychiatric morbidity, including mood disorders. Suicide attempts during adolescence are related to a 7-fold increase of the odds of a subsequent suicide attempt during young adulthood [19]. The present findings highlight that impaired emotion regulation abilities along with cyclothymic, irritable, and hyperthymic temperament differentiate young patients with a diagnosis of a mood disorder and LSI from HC. Furthermore, when directly comparing patients with and without LSI, young individuals with a diagnosis of mood disorders and LSI reported more use of lithium and antipsychotic medications, higher rates of family history of psychiatric disorders, and psychotherapy, as well as increased emotion dysregulation and cyclothymic temperament.

Contemporary conceptualizations of emotion dysregulation have moved from a dichotomous framework, in which an individual is either successful or unsuccessful at inhibiting or controlling affect, particularly negative emotions, to a multidimensional model of emotion regulation. The latter encompasses awareness, comprehension, and acceptance of emotions, the ability to engage in goal-directed behaviors, and to refrain from acting hastily when experiencing distressful affects, as well as the perception of one’s ability to effectively adopt emotion regulation strategies when situationally challenged [5]. Emotion dysregulation occurs when any of these self-regulatory domains is impaired [20]. A growing body of evidence suggests a role of emotion dysregulation in the onset and maintenance of mood disorders [21,22,23,24]. Similarly, studies examining the role of emotion dysregulation in suicide also observed that individuals who perceive themselves as incapable of exerting effective emotion regulation strategies when extremely distressed, are at increased risk of suicidal ideation, independently from mood symptoms [25,26]. According to escape theories of suicide [27], individuals wish to die when they feel overwhelmed by acute and unbearable affects that prevent them from adopting any adaptive regulation strategies. This intolerable emotional state, which is perceived as uncontrollable, leads patients to think of suicide as an effective way to escape these feelings.

While research on suicide risk has so far focused on individuals with mood disorders, the influence of temperamental features as an independent risk factor has only been partially investigated. Consistent with previous studies [13,28,29], distinct affective temperaments including cyclothymic, irritable, and hyperthymic, were associated with lifetime suicidal ideation in young patients with mood disorders. However, in our sample, only cyclothymic temperament significantly discriminated between LSI and NoLSI patients with mood disorders. Affective instability, including increased mood reactivity and lability, consistently proved to contribute to suicidal ideation [6,30]. Accordingly, cyclothymic temperament, which is characterized by abrupt shifts in mood, behavior, and rapidly changing thinking [31] may therefore represent a specific vulnerability marker for suicidal ideation in individuals with mood disorders. 

The biological underpinnings of suicidal ideation are currently unclear. However, it appears that genetic factors together with environmental influences explain most of variance [32] and may relate to decreased network strength and efficiency, which differentiate people with suicidal ideation and those free from such ideation [33]. In particular, suicidal ideation is conceived to be stress-related [34], and this is witnessed by blunted responses to dexamethasone in adolescents [35,36]. Furthermore, persons with suicidal ideation show reduced performance in emotional regulation tasks as witnessed by their inability to increase late positive potentials in response to stimuli [37]. These data point to the existence of a loop between stress, emotional dysregulation, and suicidal ideation, which matches our results.

As for the influence of affective temperaments on suicidal ideation, cyclothymic and depressive affective temperaments were found to be higher in individuals with prominent psychological distress and this effect was mediated by the lack of impulse control and lack of clarity dimensions of emotional dysregulation [38]. In another study, the cyclothymic temperament was found to be predisposed to the consequences of emotional dysregulation in an attention deficit/hyperactivity disorder [39]. Emotional dysregulation in turn was found to moderate the link between mental pain and suicidal ideation [40]. These results point to bidirectional influences between affective temperaments, emotional dysregulation, and suicidal ideation. 

Before drawing conclusions, we have to acknowledge some potential limitations. First, the cross-sectional nature of our study does not allow us to generalize our results to the entire mood disorder population and is not fit for establishing causal relationships. Second, we assessed history of LSI, using a not yet validated, semi-structured questionnaire, so it is possible that it is not sufficiently sensitive for detecting suicidality. In particular, it did not provide a quantitative measure of suicidal risk. Furthermore, we did not stratify our sample, according to mood disorder diagnosis. However, the need to split the sample according to diagnosis was set off by the fact that all included diagnoses were not in a clinically active episode. Moreover, we found no differences in belonging to the LSI or NoLSI groups among diagnoses. Finally, the reliability of self-administered questionnaires may be partially biased. On the other hand, our study has some strengths, including the investigation of the heretofore poorly investigated connection between LSI and emotional dysregulation and LSI and temperament.

In conclusion, our data highlight that temperamental features and emotion dysregulation may represent independent factors for increased vulnerability to lifetime suicidal ideation in adolescents and young adults with mood disorders while in their euthymic or “asymptomatic” phase. To confirm this association, and thus shed light on the pathway leading to suicide risk in young adults suffering from mental disorders, longitudinal studies are desirable for establishing causal relationships. 

## Figures and Tables

**Table 1 jpm-11-00865-t001:** Sociodemographic and clinical characteristics of LSI, NoLSI and HC.

Characteristics	LSI (n = 24)	NoLSI (n = 26)	HC (n = 82)	*F* or *χ*^2^	df	*p*
Age (years): mean ± (SD)	18.42 (3.61)	19.12 (3.98)	19.29 (3.90)	0.48	2	0.622
Females: n (%)	20 (83.3%)	19 (73.1%)	64 (78.0%)	0.77	2	0.682
Educational level (years): mean ± (SD)	11.75 (2.21)	11.92 (2.43)	12.33 (3.64)	0.38	2	0.686
Smokers: n (%)	9 (37.5%)	5 (19.2%)	29 (35.4%)	2.66	2	0.264
Family history of psychiatric disorders: n (%)	20 (83.3%)	15 (57.7%)	-	3.91	1	0.048
Age at onset (years): mean ± (SD)	13.75 (2.86)	14.69 (4.60)	-	0.74	1	0.394
Hospitalization: n (%)	9 (37.5%)	5 (19.2%)	-	2.07	1	0.151
Substance use: n (%)	4 (16.7%)	6 (23.1%)	-	0.32	1	0.571
Drugs:						
Antidepressants: n (%)	8 (33.3%)	8 (30.8%)	-	0.04	1	0.846
Antiepileptics: n (%)	17 (70.8%)	12 (46.2%)	-	3.12	1	0.077
Antipsychotics: n (%)	12 (50.0%)	6 (23.1%)	-	3.93	1	0.048
Lithium: n (%)	11 (45.8%)	1 (3.8%)	-	12.06	1	0.001
Benzodiazepines: n (%)	8 (33.3%)	5 (19.2%)	-	1.29	1	0.256
Diagnoses:						
Major depressive disorder: n (%)	9 (37.5%)	12 (46.2%)	-	0.83	2	0.65
Bipolar disorder: n (%)	14 (58.3%)	12 (46.2%)	-			
Persistent depressive disorder: n (%)	1 (4.2%)	2 (7.6%)	-			
Psychotherapy: n (%)	17 (70.8%)	10 (38.5%)	-	5.27	1	0.022

Abbreviations: df = degrees of freedom; HC = healthy controls; LSI = patients with lifetime suicidal ideation; NoLSI = patients without lifetime suicidal ideation; SD = standard deviation.

**Table 2 jpm-11-00865-t002:** Distribution patterns of emotional dysregulation and TEMPS-A-39 affective temperaments in LSI (N = 24), NoLSI (N = 26) and HC (N = 82).

	LSI Mean ± (SD)	NoLSI Mean ± (SD)	HC Mean ± (SD)	*F*	df	*p*	HC vs. LSI * *(p)*	HC vs. NoLSI * *(p)*	LSI vs. NoLSI * *(p)*
DERS total	89.38 (18.73)	66.73 (21.95)	64.94 (17.76)	16.07	2	<0.0001	<0.0001	1.000	<0.0001
Cyclothymic	7.79 (2.47)	5.08 (2.53)	5.10 (2.50)	11.43	2	<0.0001	<0.0001	1.000	0.001
Depressive	3.71 (1.78)	3.96 (2.07)	4.26 (2.21)	0.69	2	0.502	0.798	1.000	1.000
Irritable	6.04 (2.07)	6.12 (1.97)	4.42 (2.74)	6.82	2	0.002	0.018	0.010	1.000
Hyperthymic	5.71 (1.92)	4.92 (2.13)	4.11 (2.00)	6.38	2	0.002	0.003	0.225	0.512
Anxious	1.38 (1.17)	1.42 (0.99)	1.38 (1.10)	0.02	2	0.982	1.000	1.000	1.000

Abbreviations: DERS, Difficulties in Emotion Regulation Scale; df = Degrees of freedom; HC = Healthy controls; LSI = Patients with lifetime suicidal ideation; NoLSI = Patients without lifetime suicidal ideation; SD = Standard deviation; TEMPS-A-39 = 39-item Temperament Evaluation of Memphis, Pisa, Paris and San Diego-Autoquestionnaire. * Bonferroni post hoc test.

## Data Availability

Data will be available from the corresponding author upon reasonable request without restriction.

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
