# Peer review of "Emotional Dysregulation, Temperament and Lifetime Suicidal Ideation among Youths with Mood Disorders"

_jpm, 2021, doi:10.3390/jpm11090865_

Round 1

Reviewer 1 Report

Overall, I think it contains clinically necessary content. However, here the authors are treating patients with mood disorders as a group. However, it seems that unipolar depression and bipolar disorder in mood disorder are quite different from each other. In particular, even in the case of emotional dysregulation, it is expected to appear differently depending on different mood disorders. It is agreed that emotioanl dysregulation also occurs in unipolar depression. However, it is also true that mainly emotional dysregulation is a more commonly discussed concept in relation to bipolar disorder, especially in children and adolescents. Therefore, the author needs to clarify more clearly how many diagnoses of mood disorders are included in this study result table. In addition, it is necessary to explain the rationale for analyzing the concept of emotical dysregulation by grouping various mood disorders into one category. Such an explanation must be added to support the core conclusion of this paper that there is a link between LSI and emotional dysregulation.

Author Response

Reviewer #1

Improve Methods.

Overall, I think it contains clinically necessary content. However, here the authors are treating patients with mood disorders as a group. However, it seems that unipolar depression and bipolar disorder in mood disorder are quite different from each other. In particular, even in the case of emotional dysregulation, it is expected to appear differently depending on different mood disorders. It is agreed that emotioanl dysregulation also occurs in unipolar depression. However, it is also true that mainly emotional dysregulation is a more commonly discussed concept in relation to bipolar disorder, especially in children and adolescents. Therefore, the author needs to clarify more clearly how many diagnoses of mood disorders are included in this study result table. In addition, it is necessary to explain the rationale for analyzing the concept of emotical dysregulation by grouping various mood disorders into one category. Such an explanation must be added to support the core conclusion of this paper that there is a link between LSI and emotional dysregulation.

Response: We thank reviewer for the positive attitude towards our manuscript. We agree that MDD and bipolar disorders are different from each other, as conceptualized in the current version of the DSM-5. Nevertheless, literature indicates that both major depressive and bipolar disorder present with emotional dysregulation, which may be considered a shared features among the disorders. We elaborated on this in the Introduction and added relevant literature. We also added in Table 1 data for specific diagnoses in the lifetime suicidal ideation and no lifetime suicidal ideation groups. We tested differences in diagnoses among patients with and without suicidal ideation and we found no differences in belonging to the Lifetime suicidal ideation or No Lifetime suicidal ideation groups among diagnoses. We added a couple of sentences in limitations to support our decision to group all mood disorders in one group. We added a clarifying sentence fragment in support to your suggestion in our conclusion. We again thank Reviewer for suggestions that led us to substantially improve our manuscript.

Reviewer 2 Report

The authors investigated the involvement of emotional dysregulation and temperament in suicide risk in a sample of accurately selected young patients with mood disorders and a matched sample of healthy controls (HC). Their results have demonstrated that temperamental features, in particular cyclothymic temperament, and emotion dysregulation may represent independent factors for increased vulnerability to lifetime suicidal ideation in young adults with mood disorders.

I think the manuscript includes new and intriguing findings, however the authors should revise it according to the following concerns;

  1. The authors should discuss in detail on the biological mechanism underlying the present results., citing relevant literatures.
  2. The authors should discuss on the involvement of emotional dysregulation and temperament in suicide risk of other psychiatric disorders than mood disorders, citing relevant literatures.
                                                                                                                      1. .

Author Response

Reviewer #2

Improve Introduction, Results, and Conclusions

The authors investigated the involvement of emotional dysregulation and temperament in suicide risk in a sample of accurately selected young patients with mood disorders and a matched sample of healthy controls (HC). Their results have demonstrated that temperamental features, in particular cyclothymic temperament, and emotion dysregulation may represent independent factors for increased vulnerability to lifetime suicidal ideation in young adults with mood disorders.

We thank reviewer for carefully reading our manuscript.

I think the manuscript includes new and intriguing findings, however the authors should revise it according to the following concerns;

The authors should discuss in detail on the biological mechanism underlying the present results., citing relevant literatures.

We thank reviewer for the suggestion. Accordingly, we substantially expanded our discussion section, considering biological mechanism possibly underlying our results. We relied on relevant literature, which we found in medical databases and cited it.

The authors should discuss on the involvement of emotional dysregulation and temperament in suicide risk of other psychiatric disorders than mood disorders, citing relevant literatures.

We thank reviewer for this suggestion; we searched appropriate databases for relevant material and discussed it in the Discussion section. We thank Reviewer for providing us clues to improve our manuscript.